# sFlt-1 in Chronic Kidney Disease: Friend or Foe?

**DOI:** 10.3390/ijms232214187

**Published:** 2022-11-16

**Authors:** Masaru Matsui, Kenji Onoue, Yoshihiko Saito

**Affiliations:** 1Department of Nephrology, Nara Prefecture General Medical Center, 2-897-5 Shichijo-Nishimachi, Nara 630-8581, Japan; 2Department of Nephrology, Nara Medical University, 840 Shijo-Cho, Kashihara 634-8521, Japan; 3Department of Cardiology, Nara Medical University, 840 Shijo-Cho, Kashihara 634-8521, Japan; 4Nara Prefecture Seiwa Medical Center, 1-14-16, Mimuro, Sango-Cho, Ikoma-Gun 636-0802, Japan

**Keywords:** PlGF, sFlt-1, cardiorenal connection

## Abstract

Placental growth factor (PlGF) and its receptor, fms-like tyrosine kinase-1 (Flt-1), are important regulators involved in angiogenesis, atherogenesis, and inflammation. This review article focuses on the function of PlGF/Flt-1 signaling and its regulation by soluble Flt-1 (sFlt-1) in chronic kidney disease (CKD). Elevation of circulating sFlt-1 and downregulation of sFlt-1 in the vascular endothelium by uremic toxins and oxidative stress both exacerbate heart failure and atherosclerosis. Circulating sFlt-1 is inconsistent with sFlt-1 synthesis, because levels of matrix-bound sFlt-1 are much higher than those of circulating sFlt-1, as verified by a heparin loading test, and are drastically reduced in CKD.

## 1. Introduction

The number of patients with chronic kidney disease (CKD) has been increasing globally, and these individuals currently comprise more than 10% of the general population in developed countries [1]. CKD is associated with two critical risks: the progression of end-stage renal disease and the development of cardiovascular diseases. The death rate due to cardiovascular diseases exceeds the rate of progression to more severe nephropathy in diabetic kidney disease [2]. The coexistence of cardiovascular and renal disease is associated with a higher risk of cardiovascular events and death than the presence of either condition alone [3]. Organ cross-talk between the heart and kidneys is widely recognized, and is referred to as the “cardiorenal connection” [4]. The molecular mechanisms underlying this phenomenon are not fully understood, but the natriuretic peptide family and fibroblast growth factors have been established as leading players in the pathogenesis of cardiac hypertrophy and vascular calcification in CKD [5,6]. Recently, however, the cardiorenal connection has been shown to be increasingly complex since several underlying mechanisms have been identified; these include endothelial injury, immunological imbalance, cell death, inflammatory cascades, and oxidative stress [7]. This review focuses on a novel concept wherein the development of CKD-associated cardiovascular diseases is associated with noninfectious inflammation mediated by the signaling pathway of placental growth factor (PlGF) and its receptor, fms-like tyrosine kinase-1 (Flt-1).

## 2. PlGF/Flt-1 Pathway

### 2.1. PlGF

The human vascular endothelial growth factor (VEGF) family consists of two groups of molecules: VEGF-A, VEGF-B, and PlGF, which are primarily involved in the growth of blood vessels [8]; and VEGF-C and VEGF-D, which contribute mainly to the growth of lymphatic vessels [9]. VEGF ligands can form anti-parallel homodimers and heterodimers that optimize binding to these ligands’ preferred receptors and facilitate receptor dimerization [10].

VEGF-associated receptors (VEGFR-1, also known as Flt-1; VEGFR-2, also known as Flk-1 in mice and KDR in humans; and VEGFR-3, also known as Flt-4) are characterized by a ligand-binding region with seven immunoglobulin-like domains, a single transmembrane domain, and a tyrosine kinase domain with a long kinase insert (Figure 1). VEGF-A, VEGF-C, and VEGF-D bind to Flt-1 and Flk-1, while PlGF and VEGF-B bind only to Flt-1. The VEGF family plays a wide variety of crucial roles in the regulation of angiogenesis, lymphangiogenesis, lipid metabolism, inflammation, atherosclerosis, and tumor development [11,12].

Two years after VEGF was identified, PlGF was first discovered in the human placenta, where it is highly expressed in trophoblasts and endothelial cells and promotes placental growth [13]. PlGF was later found to be expressed in many other organs, such as the heart, lungs, and brain [14,15,16]. Unlike VEGF, PlGF is redundant for vascular development and physiological vessel maintenance in healthy adults, and it is highly expressed in pathological conditions such as malignancy, tissue ischemia, and inflammation [17]. Experimental studies showed that local delivery of the PlGF gene into the carotid artery of hypercholesterolemic rabbits enhanced intimal thickening via macrophage accumulation [18]. Moreover, treatment with a neutralizing monoclonal antibody against PlGF reduced the inflammatory cell infiltration and attenuated the development of atherosclerotic lesions in Apo E KO mice [19].

### 2.2. Flt-1 and the Soluble Isoform of Flt-1 (sFlt-1)

In 1990, a novel tyrosine kinase receptor, later known as Flt-1, was originally isolated from the human placenta by Shibuya [20]. Unlike Flk-1, the affinity of Flt-1 for its ligand, VEGF, is very high, and is at least 10-fold higher than that for VEGFR-2 [21]. However, the kinase activity of VEGFR-1 is lower, about one-tenth that of VEGFR-2 [20,22]. Flt-1 is primarily expressed on vascular endothelial cells, and it functions mainly to regulate blood vessel angiogenesis, maintenance, and permeability, as well as endothelial cell migration and proliferation [10,23,24]. Its expression on non-vascular cells further supports the non-vascular biological roles of VEGF and PlGF [25,26]. Flt-1 also contributes to the migration and activation of monocytes and macrophages, leading to inflammatory processes and angiogenesis.

In 1993, Kendall reported that a soluble isoform of Flt-1 (sFlt-1) mRNA covers six Ig regions of the full-length Flt-1, with a 31 amino acid–long tail derived from intron 13 [22]. Later investigations reported Flt-1 isoforms that are generated by alternative splicing of exons 14 and 15 [27,28] and by shedding of the Flt-1 extracellular domain [29,30]. sFlt-1 traps PlGF and VEGF-A and acts as a negative regulator of both molecules. These molecular mechanisms indicate that PlGF is pro-atherogenic, while sFlt-1, an intrinsic PlGF antagonist, is anti-atherogenic.

## 3. PlGF/sFlt-1 Imbalance in CKD

### 3.1. Circulating PlGF in CKD

Circulating levels of PlGF are normally undetectable, but increased PlGF levels have been described under pathological conditions [31]. In CKD patients, serum levels of PlGF are directly correlated with CKD severity [32] and the left ventricular mass index [33]. We found that increasing PlGF levels were associated with an increased risk of both mortality and cardiovascular events, the latter defined as atherosclerotic diseases and heart failure (HF) requiring hospitalization: the risks for patients in the highest quartile (≥19.6 pg/mL) were 3.87- and 8.42-fold higher, respectively, than for those in the lowest quartile (<10.1 pg/mL) [32]. Higher levels of PlGF and greater CKD severity were independently and additively associated with an increased risk of mortality and cardiovascular events (Figure 2).

### 3.2. PlGF Production in CKD

In a remnant kidney model, PlGF mRNA and protein expression were upregulated not only in the kidneys but also in other organs [34,35,36]. The expression of PlGF in cultured human umbilical arterial endothelial cells was also increased by the addition of human serum from patients with advanced CKD, which also upregulated biomarkers for endothelial injury and oxidative stress. However, treatment with the antioxidant vitamin E attenuated PlGF production [35].

These data suggest that upregulation of PlGF in CKD contributes to the development of arteriosclerotic diseases or HF via augmentation of vascular inflammation and permeability. Activation of mineralocorticoid receptors by aldosterone induces early atherosclerosis and promotes plaque inflammation via increased PlGF secretion from vascular cells [37]. Recently, treatment with a novel selective nonsteroidal mineralocorticoid receptor antagonist, finerenone, reduced the risks of cardiovascular events in patients with diabetic kidney disease [38,39]. This beneficial effect of finerenone in CKD may contribute to the suppression of PlGF/Flt-1 signaling.

### 3.3. Circulating sFlt-1 in CKD

Two studies in 2009 found conflicting associations between circulating sFlt-1 levels and estimated glomerular filtration rates (eGFRs). Di Marco et al. found elevated plasma sFlt-1 levels in CKD, which is consistent with the decreased eGFR in CKD and with the role of sFlt-1 as a marker of endothelial injury [40]. This result suggested that excess sFlt-1 contributes to the development of cardiovascular diseases in CKD. On the other hand, we showed that sFlt-1 levels decreased continuously with decreasing eGFR in patients who underwent cardiac catheterization, and that sFlt-1 deficiency increased the risk of atherosclerotic diseases. Subsequent studies supported the findings of Di Marco et al., demonstrating that sFlt-1 levels were higher in patients with CKD than in those without [41,42,43]. At that time, we were unable to explain this discrepancy.

A few years later, we performed new experiments in which we measured plasma sFlt-1 levels in patients who underwent a kidney biopsy. These levels were approximately one-fourth of the values of our previous measurements and were negatively correlated with eGFR, and thus the results conflicted with those in patients who underwent cardiac catheterization. To resolve this mysterious issue, we considered various possibilities such as the site of blood collection, comorbidities, and medication, and then noticed that heparin significantly altered sFlt-1 levels. Our blood samples were collected a few minutes after flushing with heparinized saline solution in catheterized patients, but heparin was not administered to patients who underwent a renal biopsy. We confirmed two critical points in the relationship between heparin and sFlt-1 levels. First, even a small amount of heparin (0.4 units/kg) markedly increased sFlt-1 levels, with greater elevation of sFlt-1 levels after higher heparin doses. Second, the increase in sFlt-1 levels after heparin administration differed drastically between patients with and without CKD. We therefore designed a new heparin loading test to understand the clinical role of sFlt-1.

### 3.4. Heparin Loading Test

In our preliminary experiments, a high heparin dose of 40 units/kg (approximately 2000 units of heparin) resulted in a significant difference in sFlt-1 plasma levels between healthy subjects and CKD patients (Figure 3A), but in healthy subjects this often resulted in peak sFlt-1 levels of over 10,000 pg/mL and high levels for more than 10 min after administration. We were concerned about the possibility of increased bleeding risk with high-dose heparin treatment, and thus we designed a heparin loading test in which plasma sFlt-1 levels were measured before and 5 min after heparin administration of 0.4 units/kg. Lower eGFR in CKD patients was associated with higher pre-heparin sFlt-1 levels but lower post-heparin sFlt-1 levels, indicating that the correlation coefficient between sFlt-1 levels and eGFR changed from negative to positive with low-dose heparin treatment **(**Figure 3B). Vascular endothelial cells can store large amounts of sFlt-1 bound to extracellular heparan sulfate [35]. Heparin releases matrix-bound sFlt-1 into the circulation by displacing the sFlt-1 heparin-binding site from heparin sulfate proteoglycans. The storage of sFlt-1 on the endothelial cell surface was confirmed by an in vitro experiment in which heparin administration released sFlt-1 from human microvascular endothelial cells without affecting sFlt-1 mRNA expression [29,30,35]. Similar results were also obtained in placental trophoblast cells [44].

### 3.5. Production of sFlt-1 in CKD

We found that sFlt-1 mRNA levels in the kidneys and lungs were decreased in 5/6-nephrectomized mice and were also decreased in cultured human endothelial cells after the addition of serum from patients with advanced CKD, but not from healthy subjects [35]. In humans, sFlt-1 mRNA levels in renal biopsy samples were lower with greater CKD severity [36]. Furthermore, sFlt-1 mRNA levels were significantly decreased in human endothelial cells incubated with the uremic toxins indoxyl sulfate and p-cresol, but these levels did not change irrespective of the addition of uremic toxins [45].

## 4. Role of PlGF/sFlt-1 Imbalance

### 4.1. Atherosclerosis in CKD Model Mice

An in vivo experiment using 5/6-nephrectomized Apo E knockout (KO) mice showed that mRNA and plasma levels of sFlt-1 were decreased but those of PlGF were increased compared with the control Apo E KO mice [35,36]. The CKD model mice had more areas with severe plaque in the thoracoabdominal aorta and aortic root, with massive infiltration of macrophages into the plaques. These atherosclerotic lesions were positively related to plasma levels of indoxyl sulfate and p-cresol [45]. Treatment with AST120, an oral carbon adsorbent of uremic toxins, was inversely related to atherosclerosis in CKD model mice and was also associated with an increase in sFlt-1 production. Additionally, repeated administration of recombinant human sFlt-1 comprising the first three immunoglobulin-like domains of the extracellular region of Flt-1 resulted in significant decreases in plaque area and macrophage infiltration (Figure 4).

### 4.2. Generation of sFlt-1 KO Mice

To confirm that the reduction in sFlt-1 alone aggravates atherosclerosis, we generated constitutive sFlt-1 KO mice in which exons 13 and 14 were directly connected and did not contain the mechanisms required for alternative splicing [35]. sFlt-1 KO mice produced full-length Flt-1 and grew normally, with no abnormalities in aortic or renal function. In sFlt-1 KO mice, sFlt-1–like immunoreactivity was present at a concentration approximately half that in wild-type mice, probably because other soluble isoforms were generated by alternative splicing or by shedding. However, the degree to which plasma sFlt-1 levels were increased by intravenous heparin administration was markedly lower in sFlt-1 KO mice than in wild-type mice. sFlt-1 Apo E double KO mice exhibited enlarged atherosclerotic plaque areas with increased macrophage infiltration, suggesting that sFlt-1 deficiency alone can induce a pro-atherogenic state in CKD.

### 4.3. Increased sFlt-1 and HF

CKD patients frequently suffer from concomitant HF, and the prevalence of HF increases with decreasing kidney function [46]. We revealed that CKD patients with high PlGF levels exhibited a high prevalence of HF requiring hospitalization [32]. Another clinical study showed that PlGF was strongly associated with elevated left ventricular mass index in patients with stage 2–4 CKD [33]. By contrast, the relationship between PlGF levels and cardiovascular events, including incident HF, lost significance when adjusted for albuminuria and eGFR in stage 2–4 CKD [43].

HF seems to be robustly linked to circulating sFlt-1 levels but not those of PlGF. sFlt-1 levels were strongly associated with adverse outcomes in patients with chronic HF [41,47,48]. CKD patients with high sFlt-1 levels had a lower left ventricular ejection fraction and experienced earlier mortality [49].

We confirmed that sFlt-1 levels did not change before or within 1 week after transverse aortic constriction surgery in HF mice [50] but another study demonstrated a significant elevation in sFlt-1 levels in similar model mice after 4 and 8 weeks of follow-up [48]. Di Marco et al. reported that rats treated with recombinant sFlt-1 showed decreased capillary density and induced myocardial interstitial fibrosis, resulting in a reduced left ventricular ejection fraction, while VEGF121, an antagonist of sFlt-1, preserved the heart microvasculature [49]. These experimental data indicate that elevated sFlt-1 levels chronically aggravate HF via microvascular endothelial injury.

### 4.4. Decreased sFlt-1 and HF

Given the downregulation of sFlt-1 mRNA expression in CKD, we investigated the relationship between sFlt-1 and HF in sFlt-1 KO mice that underwent TAC surgery as a model of CKD-associated HF. In sFlt-1 KO mice, we observed earlier onset of left ventricular hypertrophy and HF compared with wild-type mice [50]. Furthermore, sFlt-1 KO mice died significantly earlier than wild-type mice. The cause of death was determined to be HF based on increased lung weight and a reduced left ventricular ejection fraction. sFlt-1 replacement therapy prevented left ventricular hypertrophy and cardiac dysfunction in sFlt-1 KO mice with TAC, suggesting that the lack of sFlt-1 can cause HF development in mice with pressure overload.

Additionally, we investigated whether relative PlGF/Flt-1 signaling activation due to sFlt-1 downregulation contributed to the worsening of HF in this model. Anti-PlGF antibody also prevented TAC-induced remodeling and HF in the model. PlGF plays a role in macrophage mobilization and activation and stimulates the expression of several cytokines, including monocyte chemotactic protein-1 (MCP-1) [51]. In sFlt-1 KO mice with TAC, MCP-1 expression was significantly increased in ventricular myocytes and infiltrating monocytes in the interstitium. Treatment with an anti–MCP-1 antibody almost completely blocked macrophage infiltration and fibrosis formation in the ventricles and prevented the development of HF in sFlt-1 KO mice with TAC, findings that are similar to those following anti-PlGF antibody treatment. Our results show that relative activation of PlGF/Flt-1 by decreased sFlt-1 production plays a key role in the aggravation of cardiac hypertrophy and HF through upregulation of MCP-1 expression in the pressure-overloaded heart.

In a human biopsy study, dialysis patients with uremic cardiomyopathy exhibited a wide range of interstitial fibrosis as well as macrophage infiltration with higher MCP-1 expression in cardiomyocytes compared to patients with diastolic cardiomyopathy [52]. Post-heparin sFlt-1 levels in patients with uremic cardiopathy were lower than in those with diastolic cardiomyopathy. These results suggest that activated PlGF/Flt-1 signaling and subsequent macrophage-mediated chronic inflammation via MCP-1 in the myocardium are involved in the pathogenesis of uremic cardiomyopathy.

## 5. Distribution of sFlt-1

Most studies did not clearly state whether sFlt-1 was measured with or without heparin infusion. Scientists and clinicians must interpret that, in human studies, circulating sFlt-1 levels may be influenced due to heparin infusion since heparin is often preferred in patients with atherosclerosis, heart failure, and preeclampsia to prevent clotting. The heparin loading test is a diagnostic technique of understanding sFlt-1 distribution; pre-heparin sFlt-1 levels represent free sFlt-1 while post-heparin sFlt-1 levels estimate total sFlt-1 (free plus matrix-bound sFlt-1). As mentioned above, CKD presents a mixture of increased free and decreased total sFlt-1 levels. Pre-heparin sFlt-1 levels were significantly higher in HF patients than in non-HF patients, but the two groups were very similar regarding post-heparin levels, which suggests production of sFlt-1 may not be decreased in HF patients without CKD (Figure 5).

Both persistently high and low levels of sFlt-1 can be considered pathological and can result in arteriosclerosis and cardiac remodeling. These effects are due to the regulation of circulating free sFlt-1, matrix-bound sFlt-1, and total sFlt-1 (free plus matrix-bound sFlt-1) (Figure 6). CKD is characterized by the activation of a number of neurohumoral factors, including the renin–angiotensin–aldosterone system, the sympathetic nervous system, and proinflammatory cytokines such as uremic toxins, all of which lead to endothelial dysfunction [53]. Elevations of free sFlt-1 levels may result not from upregulated sFlt-1 production, but rather from the shifting of matrix-bound sFlt-1 to the circulation as a result of inflammatory endothelial injury in CKD and HF. A recent study showed that sFlt-1 mRNA expression in placental trophoblast cells was unchanged under hypoxic conditions, but sFlt-1 release into the media was significantly greater than under normoxic conditions [54]. A similar phenomenon may occur in ischemic endothelial cells in HF, given that endothelial cells, such as trophoblast cells, are a potential storage site for large amounts of sFlt-1 bound to extracellular heparan sulfate. Elevated sFlt-1 expressions in monocytes isolated from CKD patients and kidney transplantation patients were observed, but there may be other sources increasing free sFlt-1 levels [55,56]. Furthermore, total sFlt-1 is downregulated in the endothelium in CKD, which was confirmed by the finding that post-heparin sFlt-1 levels were decreased in CKD patients. Taken together, the regulation of increased free sFlt-1 levels and decreased total sFlt-1 levels can contribute to the development of atherosclerotic diseases and aggravation of HF in CKD patients.

## 6. Imbalance between PlGF and sFlt-1

There is growing evidence of interaction between PlGF and sFlt-1 in the pathophysiology of preeclampsia. The molecular characteristics of preeclampsia are logically related to an imbalance between PlGF and sFlt-1; that is, a rise in sFlt-1 and a decrease in PlGF due to placental hypoxia or impaired perfusion in the placenta trigger systemic vascular and glomerular endothelial injury and lead to the onset of clinical symptoms, including hypertension and proteinuria [57,58]. A circulating maternal sFlt-1/PlGF cut-off ratio of 38 or lower predicts a low short-term risk of developing preeclampsia [59]. By contrast, levels of both sFlt-1 and PlGF in healthy subjects are less than about one-tenth of those in pregnant women. These angiogenic factors maintain and regulate the vascular endothelial system when their ratio is appropriate, although to a lesser degree.

As schematically illustrated in Figure 7, we hypothesized that the balance between PlGF and sFlt-1 is disturbed when sFlt-1 is downregulated due to proposed triggers such as uremic toxins and reactive oxygen species in CKD patients, leading to the relative activation of atherogenic PlGF signaling through Flt-1. PlGF/Flt-1 signaling targets many cell types, including cardiomyocytes, endothelial cells, smooth muscle cells, macrophages, T cells, and fibroblasts [60]. In CKD it is likely that PlGF/Flt-1 signaling partly contributes to phenotypic changes of CKD such as ventricular hypertrophy and cardiac fibrosis with accumulation of macrophages and T cells in the heart, and smooth muscle cell proliferation and plaque formation in large arteries [61,62].

## 7. Perspectives and Conclusions

This review focused on the novel concept that abnormal distribution of sFlt-1 in the circulation and endothelium and subsequent relative activation of PlGF/Flt-1 signaling contribute to CKD-associated cardiovascular complications. Two interventional approaches for sFlt-1 levels that would improve CKD-associated cardiovascular disease should be considered. First, recombinant VEGF121 and extracorporeal removal of sFlt-1 may improve poor outcomes in order to reduce excess free sFlt-1 levels [63]. Second, the administration of sFlt-1 can rescue decreased total sFlt-1 levels and suppress PlGF/activation, but large amounts of sFlt-1 excessively suppress the beneficial physiological actions of both PlGF and VEGF, resulting in adverse effects on the heart and kidneys [49,64]. Treatment with sFlt-1 may face a host of challenges and difficulties, particularly ensuring that endogenous sFlt-1 production is maintained at physiological levels. However, increasing clinical and experimental data on PlGF/sFlt-1 signaling suggest the benefit of developing novel sFlt-1–based therapies for CKD-dependent atherosclerosis and HF.

## Figures and Tables

**Figure 1 ijms-23-14187-f001:**
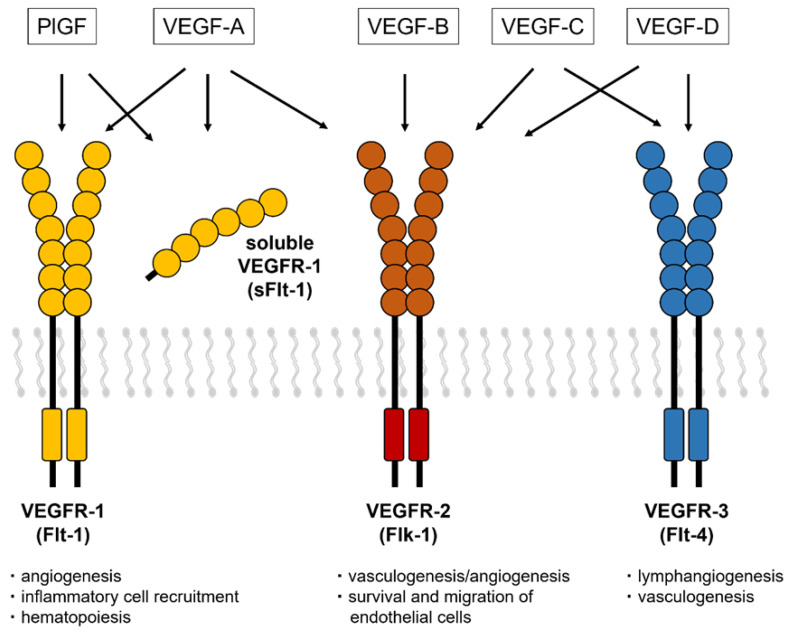
VEGF receptors and a soluble isoform of VEGFR-1. The VEGF isoforms have three receptors (VEGFR-1, VEGFR-2, and VEGFR-3) that contain a tyrosine–kinase domain and an extracellular region with seven Ig-like loops. The soluble isoform of Flt-1 consists of extracellular domains with motifs that bind to VEGF-A, PlGF, and heparin.

**Figure 2 ijms-23-14187-f002:**
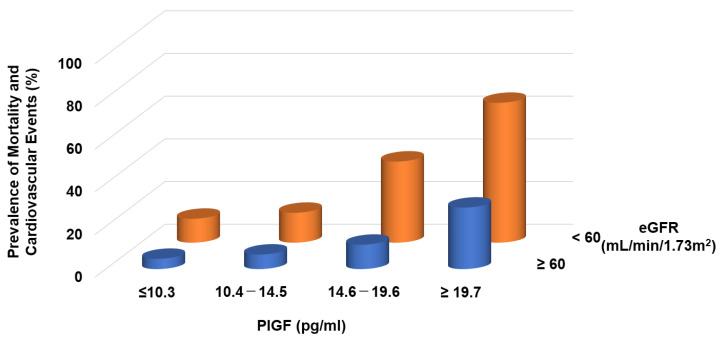
Prevalence of mortality and cardiovascular events in CKD patients. Higher PlGF levels are significantly associated with adverse events.

**Figure 3 ijms-23-14187-f003:**
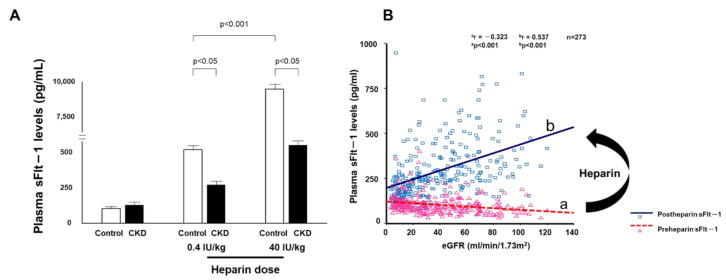
Elevated sFlt-1 with heparin treatment (**A**) Heparin reverses the relationship between sFlt-1 and eGFR. The heparin loading test shows low levels of sFlt-1 in CKD patients compared with those in control subjects at 5 min after heparin infusion. (**B**) Pre- and post-heparin sFlt-1 levels are negatively and positively correlated with eGFR, respectively.

**Figure 4 ijms-23-14187-f004:**
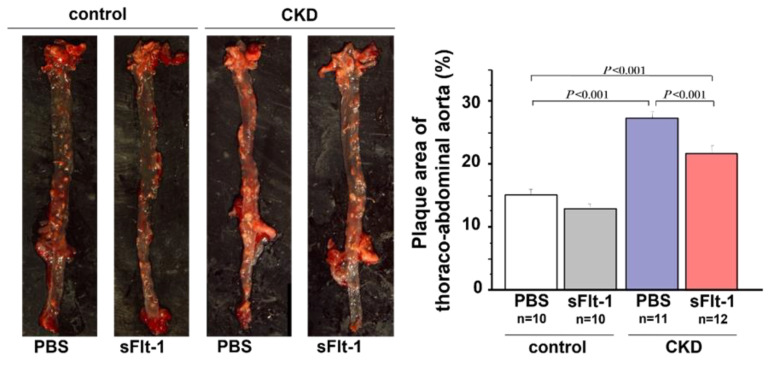
Aortic plaque formation in control or 5/6-nephrectomized Apo E KO mice. Plaque formation in the thoracoabdominal aorta was increased in a mouse remnant kidney model but reduced by repeated intraperitoneal injection of sFlt-1.

**Figure 5 ijms-23-14187-f005:**
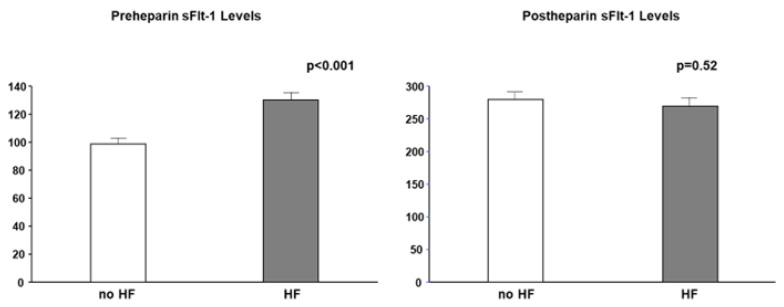
Pre- and post-heparin sFlt-1 levels in HF and no HF. Pre-heparin sFlt-1 levels are significantly increased in HF, but post-heparin sFlt-1 levels are similar between HF and no HF. HF is defined by brain natriuretic peptide ≥ 200 pg/mL.

**Figure 6 ijms-23-14187-f006:**
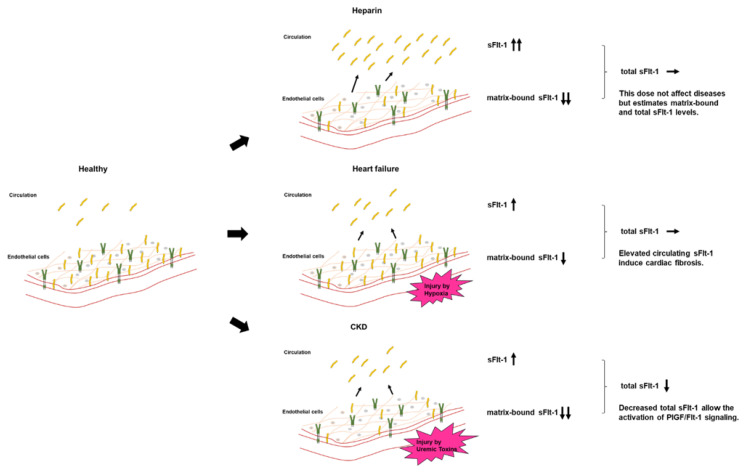
Circulating sFlt-1, matrix-bound sFlt-1, and total sFlt-1 levels in each model. Heparin infusion releases matrix-bound sFlt-1 into the circulation by displacing the sFlt-1 heparin-binding site from heparin sulfate proteoglycans. In HF, the release of free sFlt-1 from matrix-bound sFlt-1 under hypoxia may increase free sFlt-1 levels. Downregulated endothelial-derived sFlt-1 activates PlGF/Flt-1 signaling in CKD.

**Figure 7 ijms-23-14187-f007:**
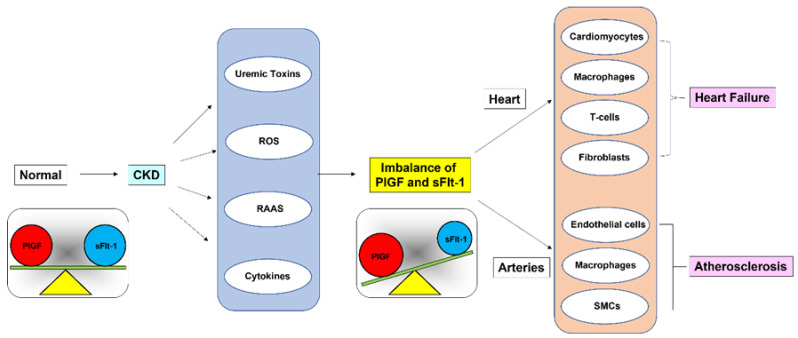
Imbalance between PlGF and sFlt-1 levels in CKD. Relative activation of PlGF/Flt-1 signaling, which is partly due to uremic toxins, exacerbates HF and contributes to plaque formation in large arteries in CKD.

## Data Availability

Not applicable.

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
