# Peer review of "sFlt-1 in Chronic Kidney Disease: Friend or Foe?"

_ijms, 2022, doi:10.3390/ijms232214187_

Round 1

Reviewer 1 Report

Matsui et al. reviewed and discussed the role of sFlt-1 in CKD. This reviewed not only summarized the research progresses, but also showed and analyzed their original data. I have some minor concerns.

1.       The data of figure 2 is original or from published article? If it is from published article, it is not suitable to show in the review.

2.       Some section titles should be more previse, such as 4.3 and 4.4

3.       Please add a perspective paragraph showing some unsolved issues nowadays and future research directions.

Reviewer 2 Report

In this manuscript Matsui et al. provide a review on the function of PIGF/Glt-1 signaling in chronic kidney disease, as well as its regulation by soluble Flt-1. The manuscript presents an interesting, well written and useful review. I think that this theme is suitable to this journal’s policy. 

My comments:

1.     The references should be up-to-date, i.e., only 29% of references in the paper have been published within recent 5 years. 

Author Response

Thank you for your useful suggestion. Our manuscript is revised with adding 6 papers within 5 years as a reference.

Reviewer 3 Report

Matsui et al present a summary of work, including some of their own, on the role of soluble FMS-like tyrosine kinase 1 in atherosclerosis associated with chronic kidney disease. They note that s-Flt1 serves as a negative regulator of VEGF and PlGF. They note that PlGF levels are elevated in CKD and independently associated with increased risk of mortality and cardiovascular events. They note that the brnificial effects of the new non-steroidal mineralocorticoid receptor blocker finerenone may be in part due to reducing PlGF/Flt-1 signaling. They site contradictory results of experiments with some showing elevated sFlt-1 levels in CKD while there own work showed a decrease in sFlt-1 levels with worsening CKD. They report that s-Flt-1 levels can be dramatically increased by heparin, which serves to unbind s-Flt-1 from matrix proteins and describe their heparin loading experiment to demonstrate that point. But they conclude that lower levels of sFlt-1 aggrevate atherosclerosis in mouse models in which s-Flt-1 in constitutively knocked out. They conclude that the uremic toxins indoxyl sulfae and p-cresol lead to decreased s-Flt-1. They cite pre-eclampsia as a model of pathology from elevated sFlt-1 levels and suggest in heart failure that hypoxia leads to release of sFlt-1 from matrix proteins. They further suggest that disturbance in the balance of PlGF/Flt-1 may be upset by reductions in sFlt-1 and speculate about a therapeutic role for sFlt-1.

I've tried to summarize the article in detail because of the complexity of the model and the frequent interjection of comments about whether experiments were conducted with or without heparin. I think the comments about heparin need to be limited to the discussion of heparin loading and perhaps summarized in a paragraph about potential confounding factors in the literature about whether heparin was used. It would be helpful to include more about the role of hypoxia in releasing sFlt-1 from matrix proteins since this seems the physiologic mechanisms.

The article needs a more robust conclusion to tie all these threads together. The final paragraph lines 302-318 looks like it could be a conclusion, but it is not identified as one.

Other questions

Line 33, what is meant by overhydration?

Line 218 sentence beginning "As verified..." doesn't make any sense. 
